# Unsupervised Meta-Learning via Dynamic Head and Heterogeneous Task Construction for Few-Shot Classification

## Abstract

Meta-learning has been widely used in recent years in areas such as few-shot learning and reinforcement learning. However, the questions of why and when it's better than other algorithms in few-shot classification remain to be explored. In this paper, we perform pre-experiments by adjusting the proportion of label noise and the degree of task heterogeneity in the dataset. We use the metric of Singular Vector Canonical Correlation Analysis to quantify the representation stability of the neural network and thus to compare the behavior of meta-learning and classical learning algorithms. We find that benefiting from the bi-level optimization strategy, the meta-learning algorithm has better robustness to label noise and heterogeneous tasks. Based on the above conclusion, we argue a promising future for meta-learning in the unsupervised area, and thus propose DHM-UHT, a dynamic head meta-learning algorithm with unsupervised heterogeneous task construction. The core idea of DHM-UHT is to use DBSCAN and dynamic head to achieve heterogeneous task construction and meta-learn the whole process of unsupervised heterogeneous task construction. On several unsupervised zero-shot and few-shot datasets, DHM-UHT obtains state-of-the-art performance. The code is released at https://github.com/tuantuange/DHM-UHT.

## 1 Introduction

Meta-learning has emerged as a powerful paradigm for learning to adapt to unseen tasks Vanschoren (2018). As an example, the optimization-based meta-learning algorithm Finn et al. (2017); Raghu et al. (2020); Nichol et al. (2018) has been shown to demonstrate excellent generalization performance in few-shot learning and reinforcement learning. In these areas, the more commonly used pre-train and fine-tune strategy exhibits disadvantages regarding training overhead, reliance on massive samples, and accuracy.

Nevertheless, in recent years, new research has shown that models pre-trained by the classical Whole-Class-Training (WCT) strategy exhibit comparable or even better accuracy on multiple few-shot image classification datasets Tian et al. (2020); Chen et al. (2021). The inconsistent conclusions described above confuse us about the nature of meta-learning, and in turn hinders us from developing the area. Most of the current theoretical researches on meta-learning focus on estimating the generalization error upper bound of meta-learning Jose & Simeone (2021a); Chen et al. (2020). However, the conclusions given by these researches cannot be directly used to improve the performance of meta-learning algorithms nor extend the range of their applications. Two simple but important questions remain to be answered – **why and when is meta-learning better than other algorithms in few-shot classification?**

The answer is that **meta-learning is more robust to label noise and heterogeneous tasks**, and that **meta-learning has better unsupervised performance under the same constraints of annotation ability**. By employing SVCCA (Singular Vector Canonical Correlation Analysis Raghu et al. (2017a)) as a metric, we propose a quantitative approach to measure representation stability. By visualizing the representation stability of each neural network layer, we can analyse the characteristics of meta-learning and other algorithms during training process. Based on the above approach, we perform pre-experiment by adjusting the proportion of label noise and the degree of task heterogeneity in the dataset. We find that compared to the models trained by classical learning algorithms, the

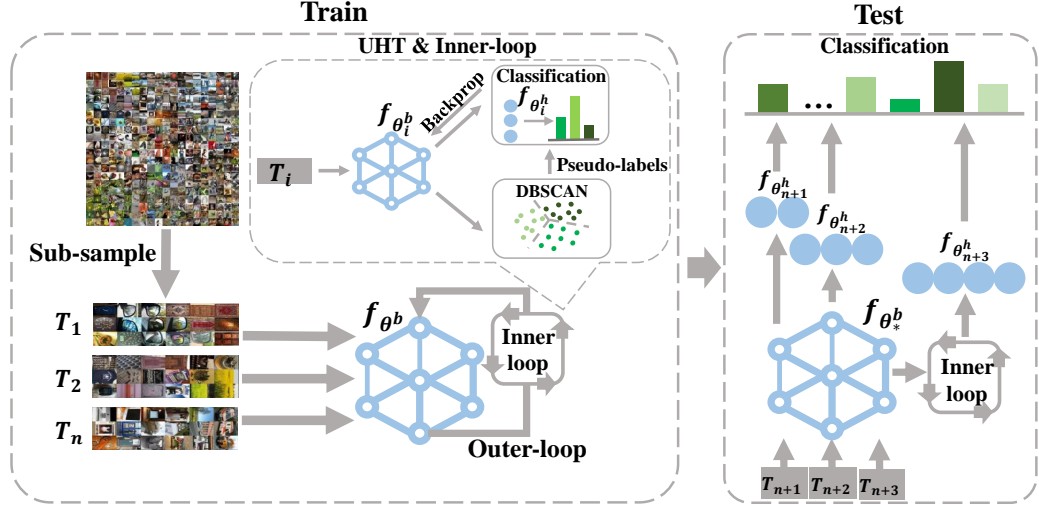

Figure 1: Overview of DHM-UHT. During the training of the left block, DHM-UHT samples unannotated datasets and then performs meta-learning based on bi-level optimization. The inner loop performs the optimization process for unsupervised heterogeneous task construction, and the outer loop learns common representations from different tasks. During testing, depending on the requirements of the few-shot and zero-shot, we can choose to perform or not to perform inner loop to fine-tune the model.

models trained by meta-learning algorithms exhibit higher representation stability in their middle layer, when there are label noise and task heterogeneous in the dataset. In terms of few-shot classification accuracy, meta-learning also outperform classical learning algorithms. The reason is, benefits from the bi-level optimization strategy, meta-learning is able to progressively learn stable representation in the middle layers (body) of the neural network while controlling unstable representation in the bottom layer (head) of the neural network. Meanwhile, previous researchers have been obsessed with static networks with identical loss functions Finn et al. (2017); Nichol et al. (2018), thus focusing only on the learning of homogeneous tasks. By improving the static head meta-learning into dynamic head meta-learning, we make the meta-learning more flexible and make it able to learn a wider range of common representation from heterogeneous tasks. In conclusion, **the advantage of meta-learning partly come from its robustness to label noise and heterogeneous tasks**, and furthermore, this robustness is derived from a more rational use of neural networks by its bi-level optimization strategy. Since meta-learning has better robustness to labeling noise and heterogeneous tasks, the converse is true: **meta-learning should have better performance in unsupervised area under the same annotation ability constraints**.

As a result, we propose a dynamic head meta-learning algorithm with unsupervised heterogeneous task construction, *i.e.*, DHM-UHT. As shown in Figure 1, the core idea of DHM-UHT is to use DBSCAN Ester et al. (1996) and dynamic head to achieve heterogeneous task construction (UHT) and **meta-learn the whole process of unsupervised heterogeneous task construction** (shown in the inner loop). Such an approach not only utilize the robustness of meta-learning in a thorough way, but also optimizes the ability to annotate. As a result it has the best performance because its optimization target is consistent with the target of downstream tasks. In contrast, most meta-learning algorithms optimize only few-shot performance.

Our contribution can be summarized as follows:

- We answer the question of why and when meta-learning is better than the classical learning algorithms in few-shot classification.

- To exploit meta-learning's robustness to label noise and heterogeneous tasks, we propose DHM-UHT. We are the first to treat the whole process of unsupervised task construction as the meta-objective.

- Through comparative experiment, ablation experiment, and sensitive experiment, we demonstrate the superiority of DHM-UHT on unsupervised zero-shot and few-shot learning scenarios.

## 2 Why and When is Meta-learning better in few-shot classification?

In this chapter we answer the two questions of why and when meta-learning is better than other algorithms in few-shot classification. We reveal that the strength of meta-learning algorithm lies in the fact that it is robust to label noise and heterogeneous tasks, and that meta-learning should have better performance in unsupervised area under the same annotation ability constraints.

### 2.1 Robustness to Label Noise

Currently, the controversy about meta-learning focuses on few-shot classification scenarios. Several studies have shown that pre-trained models obtained by classical Whole Class Training (WCT) can achieve similar or even better results than meta-learning algorithms Tian et al. (2020); Luo et al. (2023); Chen et al. (2021). However, such a straightforward comparison is unfair because under supervised conditions, the annotation effort of WCT requires distinguishing all categories in the dataset, whereas meta-learning only requires distinguishing a few categories in the meta-task. Since data annotation for classical supervised pre-training algorithm is more costly than meta-learning when obtaining similar performance, it is reasonable to speculate that meta-learning algorithm may perform better when annotation ability or label error rate are the same.

**Performance Evaluation.** First, we evaluate the classification accuracy of two meta-learning algorithms and classical WCT pre-training algorithm on Omniglot and Mini-Imagenet dataset with 5-way 1-shot task. We use the same neural network architecture and learning configuration as Finn et al. (2017); Raghu et al. (2020). The datasets setup is detailed in A.1. As shown in Table 1 and Table 2, we compared two meta-learning algorithms with WCT on the original datasets. The results are the same as in Tian et al. (2020): under the condition of using the same network architecture, there is almost no difference in performance between them. However, the meta-learning algorithms ANIL and MAML achieved significantly higher classification accuracy when we introduced 15% label noise to samples in the dataset. The difference in performance is even more pronounced when the label noise comes to 30%.

**Representation Stability Analyse.** Second, we analyse the behavior of meta-learning and WCT on neural network. Specifically, by employing SVCCA Raghu et al. (2017b) as metric, we propose a quantitative approach to measure representation stability $rs_t^i$. By visualizing the representation stability of each neural network layer, we can analyse the characteristic of meta-learning and other algorithms during training process. The learned representation of $i-th$ layer can be written as $\theta^i(D)$, where $D$ is A fixed batch of test sample. At epoch $t$, the representation stability of the $i-th$ layer of the network at is define as:

$$rs_t^i = SVCCA(\theta_t^i(D), \theta_{t-1}^i(D)). \tag{1}$$

SVCCA is the metric to measure the representation similarity. It is often used to compare the behavioral similarity of different models. In meta-learning, it is used by us to explain whether fast adaptation in MAML is essentially feature reuse Raghu et al. (2020). In this paper, we use SVCCA to calculate the representation stability of the same neural network component at different training stages, *i.e.*, $rs_t^i$. This metric helps us observe the behavior of the training algorithm at all opponents of the model during the training process. As shown in Figure 2, in Omniglot dataset with 15% noise, layers trained by WCT shows a relatively low representation stability. At the same time, we can notice that the representational stability gradually decreases from the bottom to the top layers (L4 to L0), which is consistent with the general pattern of neural networks Raghu et al. (2017b). The above phenomenon shows that WCT are difficult to learn effective representations when training with label noise. In contrast, the representation of ANIL sacrifices stability in the bottom layer (*i.e.*, head), to maintain high level representational stability on the middle layers (*i.e.*, body). We are not surprised by the lower representational stability of head. According to ANIL's principle, head's parameters are involved in the update of the inner-loop, which makes it easy to fit with different data-label dependencies for different tasks. For the body of the network, when updating within the

Table 1: Comparison of Meta-Learning's and WCT's resistance to label noise on the Omniglot dataset with 5-way 1-shot task. The metric is Accuracy%.

|  | 0% label noise | 15% label noise | 30% label noise |
|---|---|---|---|
| **WCT** | 94.51 | 82.44 | 64.65 |
| **ANIL** | 94.35 | 89.83 | 76.36 |
| **MAML** | 94.46 | 89.79 | 76.34 |

Table 2: Comparison of Meta-Learning's and WCT's resistance to label noise on the Mini-Imagenet dataset with 5-way 1-shot task. The metric is Accuracy%.

|  | 0% label noise | 15% label noise | 30% label noise |
|---|---|---|---|
| **WCT** | 47.04 | 38.92 | 29.68 |
| **ANIL** | 46.77 | 41.69 | 37.45 |
| **MAML** | 46.81 | 41.63 | 37.51 |

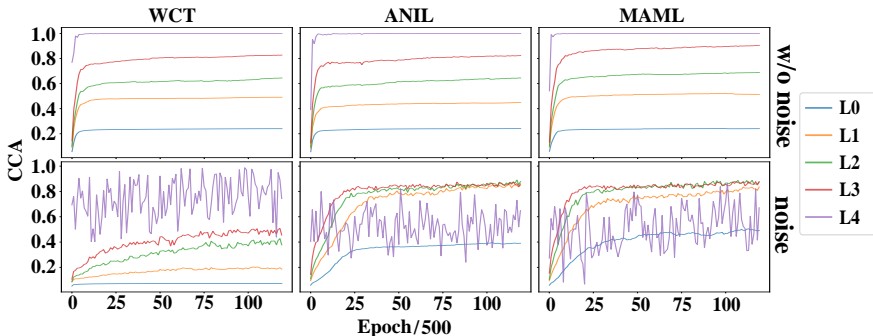

Figure 2: Representation stability of WCT and meta learning algorithms on Omniglot dataset with 5-way 1-shot tasks. The x-axis is the epoch and the y-axis is $rs_t^i$ value. Higher y-axis value means higher stability. L4 is neural network's head, L0 is the input layer, and L1-L3 is body.

outer-loop, it is able to extract inter-task common representations since heads have been adapted to the sample-label dependencies of the current tasks. **In this process, the meta-learning algorithm uses the header as a filter to avoid from label noise**. It is worth noting that MAML learns the same representation patterns as ANIL, even though it does not explicitly distinguish between head and body. We believe this is because MAML is essentially optimizing the fine-tuning process, and an good neural network pattern for fine-tuning should be the same as ANIL, MAML just find it! Back to WCT, **WCT seems to treat the whole neural network as a head**, and thus may suffer from different sample-data dependencies.

## 2.2 ROBUSTNESS TO HETEROGENEOUS TASK

In the study of meta-learning, previous researchers were obsessed with fixed networks and fixed loss functions, thus focusing only on the learning of homogeneous tasks. For example, for classification tasks, previous researchers constructed tasks with the same classification way. This approach allows different tasks to use the same network structure and identical function, which in turn simplify the algorithm and save overhead. **However, such an approach may make meta-learning overfit to unnecessary information about "way of classification" on training sets with homogeneous tasks, and underperform on test sets with heterogeneous tasks**.

To extend the heterogeneous adaptability of meta-learning, we combine the bi-level optimization training strategy of meta-learning and the dynamic head trick of multi-task learning to achieve dynamic head meta-learning (DHM). Specifically, as shown in Algorithm 1, we improve the operations related to the network's head. For each task $T_i$, we reinitialize the network's head. We will show experimentally that dynamic meta-learning can provide better generalization without loss of accuracy compared to classical multi-task learning and classical meta-learning algorithms.

In datasets composed of heterogeneous tasks, we compare the performance of dynamic head meta-learning, classical static head meta-learning (SHM), and multi-task learning (MTL). Specifically, we perform the experiment on Omniglot and Mini-Imagenet datasets. On Omniglot dataset, the classification way of heterogeneous tasks are vary from 5-20, and this number on Mini-Imagenet is 5-10. The setup is detailed in A. For SHM, we train a model for each way of tasks, and ultimately taking the average testing performance of the models. For DHM and MTL, we train the model with

Table 3: Comparison of Accuracy% on Heterogeneous Tasks.

|  | Omniglot | Mini-Imagenet |
|---|---|---|
| **Dynamic Head** | 93.27 | 44.09 |
| **Static Head** | 92.86 | 41.63 |
| **Multi Task** | 72.95 | 35.86 |

Table 4: Comparison of Accuracy% on 5-way 1-shot Homogeneous Tasks.

|  | Omniglot | Mini-Imagenet |
|---|---|---|
| **Dynamic Head** | 94.41 | 46.77 |
| **Static Head** | 94.46 | 46.81 |

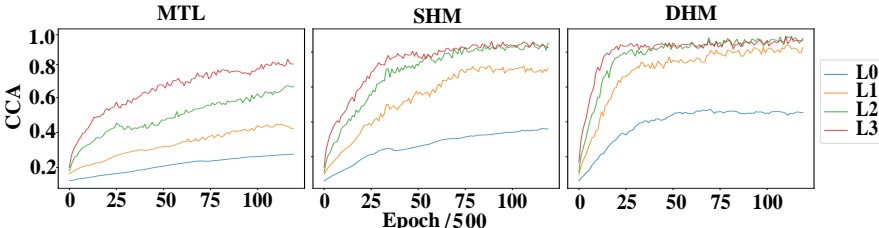

Figure 3: Representation stability of DHM, SHM, and MTL on Omniglot dataset with 5-way 1-shot tasks.

the train set consisting of a mixture of the heterogeneous tasks, and ultimately evaluating its performance directly on the test set. As shown in Table 3, we can find that the accuracy in descending is DHM, SHM, and classical MTL. Among them, DHM and SHM perform similarly and much better than MTL. Note that the degree of heterogeneity of Omniglot is higher than Mini-Imagenet, so the performance gap between each algorithms is much larger on Omniglot datasets. This result demonstrates the robustness of DHM for heterogeneous tasks. In addition, we compare the performance of DHM with SHM learning on homogeneous 5-way 1-shot test tasks. We use the same raw data for both of them. We construct heterogeneous tasks for DHM and construct 5-way 1-shot homogeneous tasks for SHM. As shown in Table 4, the performance of the DHM is comparable to SHM, with a accuracy difference at most 0.05%.

To further support the above experimental result, we also visualize the representation stability of the above three models under the task heterogeneous condition. As shown in Figure 3, during the learning process, the body of DHM obtains the most stable representation.

## 3   DHM-UHT

Since meta-learning has better robustness to label noise and heterogeneous tasks, the converse is true: meta-learning should have better performance under the same unsupervised annotation capacity constraints. As a result, we propose DHM-UHT, a dynamic head meta-learning algorithm with unsupervised heterogeneous task construction. The overview of the proposed method is shown in Figure 1. The core idea of our method is to meta-learn the process of Unsupervised Heterogeneous Task Construction. In other words, we put the process of UHT in the inner-loop. In addition, in order to accommodate and utilize heterogeneous tasks construct by DBSCAN, we re-initialize a dynamic head for each task. Note that UHT is similar to DeepCluster, with the difference that we substitute K-means to DBSCAN there. To avoid overfitting to sampling noise, we dropout the cluster with relatively small scale, when the training epoch exceeds a certain threshold (hyperparameter "min_sample" in DBSCAN). We have two reasons to use DBSCAN. On the one hand, DBSCAN has higher flexibility compared to K-means due to the unfixed number of clusters. On the other hand, learning on the heterogeneous tasks constructed by DBSCAN effectively exploits the robustness of meta-learning to heterogeneous.

**Training and Testing.** During training phase, by sampling the dataset $D$, we obtain $\{T_1, T_2, ..., T_n\}$ as the input of the meta-learning model $f_{\theta^b}$. For each $T_i$, we copy a body $f_{\theta^b_i}$ from $f_{\theta^b}$ and initialize a head $f_{\theta^h_i}$ to learn the task in an unsupervised manner. When all the inner loop and all the task $T_i$

are learned [1], we perform update to optimize $f_{\theta^b}$ and finish one step of outer loop. The optimization problem corresponding to train $f_{\theta^b}$ can be written as:

$$\arg\min_{\theta^b} \sum_{T_i \in D} L_{UHT}(f_{\theta^b}, T_i), \tag{2}$$

and one iterative solution can be written as:

$$f_{\theta^b} = f_{\theta^b} - \eta \nabla \sum_{T_i \in D} L_{UHT}(f_{\theta^b}, T_i). \tag{3}$$

When all outer loops are finished, we obtain a well trained neural network $f_{\theta^b}$. During testing phase, our method can evaluate in an unsupervised manner. Depending on the requirements of the few-shot and zero-shot, we can choose to perform or not to perform inner loop to fine-tune $f_{\theta^b}^*$.

**Unsupervised Heterogeneous Task Construction – UHT.** The process of UHT is shown in the bubble frame. For the samples in $T_i$, we use $f_{\theta^b}$ to project them in a embedding space, and then use DBSCAN to divide the embedding representation into multiple clusters. Since meta-learning model don't care the data-label dependency category Yin et al. (2020), we take the serial number of the cluster as the samples pseudo-labels. Finally, to achieve gradient computation and back-propagation, we initialize a fully connect layer as the head of neural network, and use CrossEntropy (CE) as loss function. Now we have sample, label, loss function and meta-objective Hospedales et al. (2021) (*i.e.*, the process of UHT), so we can perform inner-loop to update $f_{\theta_i}$ and outer-loop to update $f_{\theta^b}$. The loss used in the outer loop, *i.e.*, loss of the process of UHT can be written as:

$$L_{UHT_i} = L_{inner}(f_{\theta_i'}, T_i) \tag{4}$$

where $f_{\theta_i} = f_{\theta_i^b} \circ f_{\theta_i^h}$, and $f_{\theta_i'}$ is the updated base learner:

$$f_{\theta_i'} = f_{\theta_i} - \alpha \nabla L_{inner}(f_{\theta_i}, T_i), \tag{5}$$

where the loss of inner loop can be written as:

$$L_{inner}(f_{\theta_i}, T_i) = \sum_{x_i \in T_i} CE(f_{\theta_i^b} \circ f_{\theta_i^h}(x_i), f_c \circ f_{\theta_i^h}(x_i)). \tag{6}$$

Note that we use a MAML-style update strategy for base learner (*i.e.*, Equation 5), for efficiency reasons, we can also use an ANIL-style update strategy. We will compare these two approaches in Section 4.4. Also note that during the above process, we can utilize the classical support set & query set split to calculate $L_{UHT}$ and $L_{inner}$, or just use $T_i$ to calculate both of them Nichol et al. (2018).

**Dynamic Head Meta-Learning – DHM.** In the meta-learning phase, we use gradient base meta-learning as training framework, *e.g.*, MAML and ANIL. As shown in the right part of Figure 1, we divide the neural network $\theta$ to two part, body $\theta^b$ and head $\theta^h$. The body $\theta^b$ is fixed and can be shared by each task. The heads $\theta^h$ are dynamic and are customized to different tasks. In the scenarios of few-shot classification, the heterogeneity among tasks come from the difference in the number of classification ways, so for these heterogeneous tasks we use fully connected layers as heads with different length. For the same reason as DBSCAN, we use dynamic head here is to accommodate heterogeneous tasks and to exploit the robustness of meta-learning to heterogeneous tasks.

It is important to state here that our approach is fundamentally different from other unsupervised meta-learning algorithms. **DHM-UHT are the first to treat the whole process of unsupervised heterogeneous task construction as a meta-object for meta-learning, which implies it directly optimized the ability to annotation**. Other methods, represented by the CACTUs, use pre-trained feature extractors that are not trained by means of meta-learning. This implies, that such training method cannot optimize the process of generate pseudo-labels, cannot optimize test objective directly, and thus may perform sub-optimally. The process of DHM-UHT is is outlined in Algorithm 1 (non meta-batch version).

---

[1]For convenience, we roughly take a batch of data $T_i$ as a task.

**Algorithm 1** Dynamic Meta-Learning

---

1: **Require:** Dataset $D$; Neural network body $f_{\theta^b}$; Cluster algorithm $f_c$.
2: **while** not done **do**
3:    **for** $T_i \in D$ **do**
4:       **for** Inner loop **do**
5:          Initialize head: $f_{\theta_i^h}$
6:          $f_{\theta_i^b} \leftarrow f_{\theta^b}$
7:          $L_{inner}(f_{\theta_i}, T_i) \leftarrow Equation\ 6$
8:          $f_{\theta_i'} \leftarrow f_{\theta_i} - \alpha \nabla L_{inner}(f_{\theta_i}, T_i)$
9:       **end for**
10:      $L_{UHT_i} \leftarrow L_{inner}(f_{\theta_i'}, T_i)$
11:    **end for**
12:    $f_{\theta^b} \leftarrow f_{\theta^b} - \eta \nabla \sum_{T_i \in D} L_{UHT}(f_{\theta^b}, T_i)$
13: **end while**

---

Figure 4: Unsupervised Datasets Description

| Datasets | Testing Task |
|---|---|
| Cifar10 | 10w-0s |
| Cifar100 | 100w-0s |
| STL10 | 10w-0s |
| ImageNet | 1000w-0s |
| Tiny-Imagenet | 200w-0s |
| DomainNet | 345w-0s |
| Omniglot | 5w-1s/ 5w-5s/ 20w-1s/ 20w-5s |
| Mini-Imagenet | 5w-1s/ 5w-5s/ 5w-20s/ 5w-50s |

# 4 EXPERIMENT

In this section, we answer the following questions:

- Is DHM-UHT superior compared to the mainstream unsupervised few-shot and zero-shot classification algorithms?
- How effective are the components of DHM-UHT?
- How sensitive is DHM-UHT to newly introduced hyperparameters?

All our experiment results are mean values from five replicate experiments.

## 4.1 UNSUPERVISED ZERO-SHOT CLASSIFICATION

We compare DHM-UHT with several unsupervised representation learning algorithms, *i.e.*, ReSSL Zheng et al. (2022), IIC Ji et al. (2019), DeepCluster Caron et al. (2018), BiGAN Donahue et al. (2017b), MAE He et al. (2022), and NVAE Vahdat & Kautz (2020). For all of these algorithms, we use K-means to perform downstream classification. As shown in Table 4, to demonstrate the unsupervised zero-shot classification ability of DHM-UHT, we compare the models on Cifar10, Cifar100, STL10, Imagenet, Tiny-Imagenet, and DomainNet datasets. The dataset and algorithm setup is detailed in Appendix A.

Table 5 shows the accuracy of each models on the six datasets. Compared to state-of-the-art methods ReSSL, DHM-UHT obtains a higher accuracy of 5.37% on average. One of the obvious comparisons is between DHM-UHT, DeepCluster, and IIC. All three methods use a clustering-based classification strategy during training phase, however Deepcluster and IIC do not perform the meta-learning process, and thus cannot directly optimize for the ability to perform a few-shot, let alone a zero-shot learning. As a result, they may be more susceptible to biased interference from samples in a zero-shot scenario. MAE, NVAE, and BiGAN exhibit similar performance. They are essentially structured as auto-encoders that learn features indirectly by learning reconstruction process. In contrast to DHM-UHT, they are not only unable to optimize directly for downstream task objectives, but also unable to obtain resistance to noise from few-shot or zero-shot setting. ReSSL is a relation-based self-supervised algorithm, since it focuses on the relationships between different instances rather than instance level information. Similar to the Siamese network in few-shot learning, it achieves sub-optimal performance here.

## 4.2 UNSUPERVISED FEW-SHOT CLASSIFICATION

We compare our DHM-UHT with several state-of-the-art few-shot unsupervised meta-learning classification algorithms, *i.e.*, CACTUs Hsu et al. (2019), UMTRA Khodadadeh et al. (2019), Meta-GMVAE Lee et al. (2021), and PsCo Jang et al. (2023a). Note that by varying the meta-learning

Table 5: Accuracy in % on unsupervised zero-shot scenario

|  | CIFAR-10 | CIFAR-100 | STL-10 | ImageNet | Tiny-MINIST | DomainNet |
|---|---|---|---|---|---|---|
| **DHM-UHT** | 72.15 ± 1.09 | 42.34 ± 1.41 | 59.74 ± 1.35 | 30.12 ± 1.06 | 85.45 ± 1.28 | 21.68 ± 0.91 |
| **ReSSL** | 70.27 ± 1.28 | 41.48 ± 1.60 | 58.52 ± 1.31 | 31.25 ± 1.13 | 83.17 ± 1.24 | 21.42 ± 0.92 |
| **IIC** | 64.05 ± 1.02 | 36.23 ± 1.27 | 53.78 ± 1.30 | 25.07 ± 0.88 | 79.21 ± 1.54 | 18.18 ± 0.74 |
| **MAE** | 68.83 ± 1.19 | 39.11 ± 1.52 | 56.19 ± 1.47 | 27.32 ± 1.14 | 81.03 ± 1.36 | 20.53 ± 1.03 |
| **NVAE** | 67.43 ± 1.37 | 38.29 ± 1.45 | 55.78 ± 1.22 | 27.21 ± 0.98 | 81.52 ± 1.61 | 19.84 ± 0.79 |
| **DeepCluster** | 63.02 ± 1.14 | 35.05 ± 1.11 | 52.21 ± 1.42 | 24.83 ± 0.95 | 78.63 ± 1.68 | 18.09 ± 0.88 |
| **BiGAN** | 67.61 ± 1.24 | 38.78 ± 1.19 | 55.24 ± 1.34 | 26.85 ± 1.07 | 80.09 ± 1.27 | 19.23 ± 0.95 |

Table 6: Accuracy in % on unsupervised few-shot scenario

|  | Omniglot | | | | Mini-Imagenet | | | |
|---|---|---|---|---|---|---|---|---|
| (way, shot) | (5, 1) | (5, 5) | (20, 1) | (20, 5) | (5, 1) | (5, 5) | (5, 20) | (5, 50) |
| **DHM-UHT** | 93.75 ± 0.46 | 97.71 ± 0.37 | 82.15 ± 0.41 | 91.88 ± 0.40 | 44.73 ± 1.01 | 56.54 ± 0.78 | 67.30 ± 0.95 | 70.23 ± 1.07 |
| **PsCo** | 93.25 ± 0.59 | 97.56 ± 0.34 | 82.06 ± 0.43 | 91.01 ± 0.45 | 42.90 ± 0.95 | 54.87 ± 0.94 | 65.66 ± 1.05 | 69.94 ± 1.11 |
| **Meta-GMVAE** | 93.81 ± 0.75 | 96.85 ± 0.50 | 81.29 ± 0.62 | 89.00 ± 0.51 | 41.78 ± 1.13 | 54.15 ± 0.87 | 62.11 ± 1.14 | 67.11 ± 1.10 |
| **UMTRA** | 82.97 ± 0.68 | 94.84 ± 0.60 | 73.51 ± 0.53 | 91.22 ± 0.59 | 39.14 ± 1.02 | 49.21 ± 0.90 | 57.66 ± 1.02 | 59.68 ± 1.17 |
| **CACTUs-MA-DC** | 67.98 ± 0.80 | 87.07 ± 0.63 | 47.48 ± 0.59 | 72.21 ± 0.54 | 39.11 ± 1.08 | 53.40 ± 0.88 | 63.00 ± 1.06 | 68.62 ± 1.12 |
| **CACTUs-Pr-DC** | 67.08 ± 0.72 | 82.97 ± 0.64 | 46.32 ± 0.51 | 65.75 ± 0.62 | 38.47 ± 1.14 | 53.01 ± 0.91 | 61.05 ± 1.09 | 62.82 ± 1.08 |
| **CACTUs-MA-Bi** | 57.84 ± 0.75 | 78.12 ± 0.67 | 34.98 ± 0.57 | 57.75 ± 0.58 | 36.13 ± 1.07 | 50.45 ± 0.90 | 60.97 ± 1.16 | 66.34 ± 1.14 |
| **CACTUs-Pr-Bi** | 53.58 ± 0.65 | 71.21 ± 0.68 | 32.79 ± 0.53 | 50.12 ± 0.51 | 36.05 ± 1.06 | 49.87 ± 0.92 | 58.47 ± 0.09 | 62.56 ± 1.10 |
| **MAML (oracle)** | 94.46 | 98.83 | 84.6 | 96.29 | 46.81 | 62.13 | 71.03 | 75.54 |
| **ProtoNets (oracle)** | 98.35 | 99.58 | 95.31 | 98.81 | 46.56 | 62.29 | 70.05 | 72.04 |

training approach and the unsupervised embedding algorithm, there are four implementations of CACTUs, *i.e.*, CACTUs-MA-DC, CACTUs-Pr-DC, CACTUs-MA-Bi, and CACTUs-Pr-Bi. Pr represents ProtoNet Snell et al. (2017), MA represents MAML, DC represents DeepCluster, Bi represents BiGAN. As shown in Table 4, we evaluate the models on Omniglot and Mini-Imagenet datasets with several few-shot tasks. The datasets and algorithm setup is detailed in Appendix A.

Table 6 shows the accuracy of each models on the two datasets. Comparing to state-of-the-art algorithm, DHM-UHT obtains a higher accuracy of 3.52% on average. It can be seen that none of other algorithms include the process of unsupervised task construction in inner-loop, thus cannot optimize the ability of pseudo labels annotation. Among them, PsCo obtains relatively good performance due to its use of Pseudo-supervised Contrast, to target the meta-learning reliance on the immutable pseudo-labels. Note that UMTRA obtains the worst results on the Mini-Imagenet dataset. The reason is that UMTRA essentially only implements the 1-shot task construct during training. Although it uses various data augmentation methods to increase the number of shots in the constructed task, it still suffers from overfitting. We can find that the difference in performance between DHM-UHT and UMTRA increases as the number of shots increases. On the other hand, since UMTRA underlies a statistical assumption on sample labeling, its performance degrades rapidly due to mislabel in scenarios where the total number of classes is small and the number of classes in a single task is big.

## 4.3 ABLATION STUDY

We perform ablation experiment on both unsupervised few-shot and zero shot datasets, *i.e.*, Omniglot, Cifar100, and STL10 datasets. In the first control group (G1), we use K-means to generate homogeneous tasks and use static head to learn these tasks. In the second group (G2), we don't meta-learn the whole process of unsupervised heterogeneous tasks construction, instead, we generate pseudo-labels in an untrainable way, just like CACTUs. In the third group (G3), we replace dynamic head by static head, while to ensure fair comparisons we still use DBSCAN. We resample tasks with an uncertain number of ways generated by DBSCAN to generate homogeneous tasks with a fixed number of ways. We perform ablation experiment on both zero-shot and few-shot scenario. The dataset and algorithm setup are the same as the above section. As shown in Table 7, the best performance of the DHM-UHT demonstrate the necessity of DBSCAN, dynamic head, and meta-learn the whole process of UHT.

In addition, we visualized the cluster results given by DBSCAN and K-means. To obtain the scatters in Figure 5, we collect the features output by $f_{\theta^b}$, and then perform T-SNE Van der Maaten & Hinton

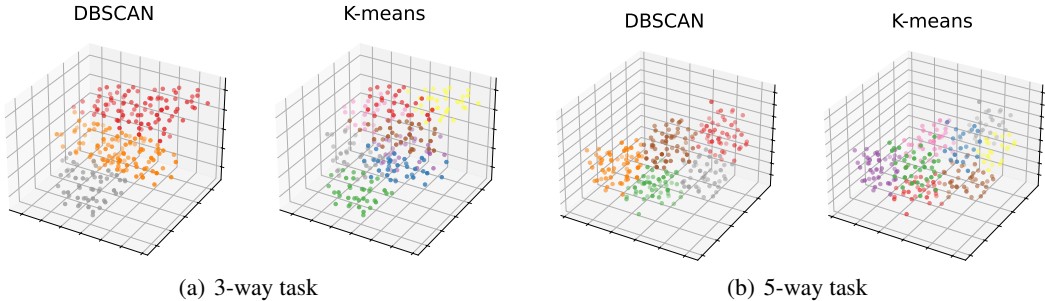

| DBSCAN | K-means | DBSCAN | K-means |
|---|---|---|---|
| (a) 3-way task | | (b) 5-way task | |

Figure 5: Clustering Visualization of DBSCAN and K-means.

Table 7: Effectiveness of each component. We compared the classification Accuracy% on the unsupervised few-shot and zero-shot scenario.

| | Omniglot | | | | Cifar100 | STL10 |
|---|---|---|---|---|---|---|
| (w, s) | (5, 1) | (5, 5) | (20, 1) | (20, 5) | (100,0) | (10,0) |
| G1 | 87.12 | 92.67 | 73.49 | 80.73 | 37.58 | 52.27 |
| G2 | 74.32 | 90.91 | 51.83 | 77.42 | 32.37 | 47.75 |
| G3 | 91.56 | 94.12 | 79.68 | 87.21 | 40.19 | 56.84 |
| **Ours** | **93.81** | **96.85** | **81.29** | **89.00** | **42.34** | **58.74** |

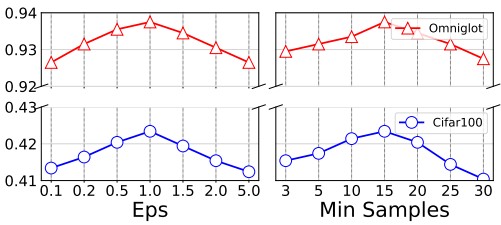

Figure 6: Sensitivity of DHM-UHT to new hyperparameters on unsupervised learning and unsupervised few-shot learning datasets.

(2008) to reduce their dimension and visualize the three dimensions scatters. We input the unlabelled sample from 3-way and 5-way tasks into our backbone trained by Omniglot dataset. It's obvious that DBSCAN can better distinguish different categories from each other, and can also better label same category scatters together. This result shows in Figure demonstrate that using DBSCAN can separate samples appropriately. while using K-means raises the problem of over-segmentation.

## 4.4 SENSITIVITY STUDY

**Hyperparameters selection.** We perform sensitivity experiment on omniglot and Cifar100 datasets. It's well known that meta-learning methods and unsupervised methods struggled with hyperparameter tuning, however, our proposed method only introduces few hyperparameters compared to CAC-TUs. With most parameter settings consistent with CACTUs, we analyze the impact of the *eps* (scanning radius of DBSCAN) and *min_samples* (minimum number of samples within a task) on the overall performance. Figure 6 shows the result evaluated on omniglot and Cifar100 datasets. We can find that it is easy for DHM-UHT to obtain a relatively high and stable performance when eps takes values in the range 0.1-5.0 and *min_samples* takes a value in the range 3-30. This result shows that our method is not particularly sensitive to the new hyperparameters thus is highly adaptable in practical applications.

**The update strategy study.** To discuss the impact of the MAML-style and ANIL-style update strategy on the UHT in terms of computational overhead and Accuracy, we recorded the computational time required for both strategy to reach the same classification accuracy. As shown in Table 8, on Omniglot dataset with 5-way 1-shot task, UHT updated by ANIL strategy can be calculated at a much faster rate. However, UHT-MAML can reach higher Accuracy ultimately. Therefore, in practice, we need to choose the update strategy based on the accuracy and overhead requirements. Note that the above results are somewhat different from those in Raghu et al. (2020), which may be due to the complexity of the unsupervised task. Updating only the head in the inner loop cannot adequately learn the process of unsupervised task construction. In scenarios where absolute high accuracy is desirable, it seems more effective to use a MAML-style update strategy.

Table 8: Comparison of UHT-MAML and UHT-ANIL on performance-overhead trade off. We recorded the computational time (second) required for both strategy to reach the same classification accuracy, on Omniglot dataset with 5-way 1-shot setting.

| Accuracy% | UHT-ANIL | UHT-MAML |
|---|---|---|
| 50.0 | 655 | 748 |
| 60.0 | 1429 | 1681 |
| 70.0 | 2730 | 2992 |
| 80.0 | 5564 | 5642 |
| 90.0 | 13987 | 14723 |
| 93.5 | / | 25717 |

## 5 RELATED WORK

**Theoretical analysis of meta-learning** The exploration of meta-learning theories has progressed in the last few years. For example, Raghu et al. (2020); Goldblum et al. (2020) unrevealed the nature of MAML's fast adaption, Tian et al. (2020) argue that learning a good embedding may outperforms meta-learning in few-shot classification scenario. Luo et al. (2023) empirically proved that meta-training algorithm and the adaptation algorithm can be completely disentangled. Chen et al. (2019) found that baseline can outperform meta-learning in the area of few-shot classification under specific conditions of domain difference and backbone network architecture. Chen et al. (2021) discuss the effect of class generalization and novel class generalization on meta-learning. Guan & Lu (2022); Jose & Simeone (2021b); Maurer (2005) estimated the generalization error upper bound of meta-learning. In this paper, we answer two simple but important questions – why and when meta-learning is better than other algorithms in few-shot classification?

**Unsupervised meta-learning.** Unsupervised meta-learning Hsu et al. (2019); Khodadadeh et al. (2019); Lee et al. (2021); Jang et al. (2023a); Ye et al. (2023); Jang et al. (2023b); Lee et al. (2023); Dong et al. (2022); Khodadadeh et al. (2021) aims to link meta-learning and unsupervised learning by constructing synthetic tasks and extracting the meaningful information from unlabeled data. Our proposed DHM-UHT is the first algorithm to meta-learn the whole process of heterogeneous unsupervised task construction.

## 6 CONCLUSION

In this paper, we answer the question of why and when meta-learning is better than classical learning algorithm in few-shot classification. The answer is that meta-learning is more robust to label noise and heterogeneous tasks, and that meta-learning has better unsupervised performance under the same constraints of annotation ability. We propose a quantitative approach to measure representation stability, and further to analyse the manner of meta-learning and other learning algorithms during training process. In the pre-experiment we find that meta-learning algorithm is more robust to label noise and task heterogeneous, cause it can train neural network in a more rational way, *i.e.*, bi-level optimization. To utilize the robustness of meta-learning, we propose DHM-UHT, a dynamic head meta-learning algorithm with unsupervised heterogeneous task construction. It's the first meta-learning algorithm treat the whole process of unsupervised heterogeneous task construction as meta-objective, and exhibit state-of-the-art performance on unsupervised zero-shot and few-shot datasets.

## REPRODUCIBILITY STATEMENT

In Sections 4 and Appendix A, we provide the necessary experimental details. Additionally, our code is released at  https://github.com/tuantuange/DHM-UHT..

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

## A  APPENDIX

### A.1  DATASET SETUP

**Omniglot.** The raw dataset contains 1628 classes, we split the classes of training set, evaluation set, test set into 800: 400: 432. We use Omniglot in three scenario. The first scenario is in 2.1. We perform supervised few-shot learning with label noise. We randomly mask the labels of the samples in the training set according to the noise ratio (*i.e.*, 0%, 15%, 30%). Depending on the training method, we can construct these raw data into task by Finn et al. (2017), or use them directly for whole class training by Tian et al. (2020). The second scenario is in Section 2.2. We perform supervised few-shot learning with heterogeneous tasks. When constructing heterogeneous tasks, we sample a variable number of classes, to ensure the difference in the way of tasks (*i.e.*, 5-20 way), and further to ensure the heterogeneity. The third scenario is in 4.2. We perform unsupervised few-shot learning. We follow the protocol given by Hsu et al. (2019).

**Mini-Imagenet.** The raw Mini-Imagenet contains 100 classes, we the split classes of training set, evaluation set, test set into 64: 16: 20. We use Mini-Imagenet in three scenario. The details of

the setup of the three experimental scenarios are the same as Omniglot. With the except that we construct 5-10 way heterogeneous task in Section 2.2.

**CIFAR-10, CIFAR-100, STL-10, Imagennet, and Tiny Imagenet.** For CIFAR-10, CIFAR-100, STL-10, Imagennet, and Tiny Imagenet datasets, we follow the protocol given by Zheng et al. (2022). They are used for unsupervised zero-shot learning, so we mask all the labels in training set.

**DomainNet.** DomainNet is a domain adaption dataset. We use it to evaluate algorithms' ability of unsupervised zero-shot domain adaption. It contains 6 domain with 345 classes for each domain. We use one domain for test and the remain 5 domain for both training and validating. Note that when constructing tasks, we sample classes from the same domain and we mask all the labels in training set.

## A.2 Algorithm Setup

**DHM-UHT** We use DHM-UHT in both unsupervised zero-shot and few-shot scenario. In unsupervised few-shot datasets, we follow the same backbone architecture given by github.com/dragen1860/MAML-Pytorch. We set epoch, inner-loop learning rate, outer-loop learning rate, meta-batch size, adaption steps for evaluation and sub-sample size, as 30000, 0.05, 0.001, 8, 50, and 100 respectively. For DBSCAN in UHT, we set min_samples and eps as 15 and 1.0, respectively. In unsupervised zero-shot datasets (except of DomainNet), we follow the same backbone architecture given by github.com/xu-ji/IIC, *i.e.*, ResNet and VGG11. We set epoch, inner-loop learning rate, outer-loop learning rate, meta-batch size, adaption steps for evaluation and sub-sample size, as 80000, 0.001, 0.001, 8, 0, and 100 respectively. For DBSCAN in UHT, we set min_samples and eps as 15 and 1.0, respectively. For DomainNet dataset, we use ResNet-9 as backbone architecture, which is the same as github.com/liyunsheng13/DRT. The other configuration is the same as other unsupervised zero-shot datasets.

**WCT, MAML, ANIL, and MTL** In Section 2, we use WCT, MAML, ANIL, and MTL to perform pre-experiment, on Omniglot and Mini-Imagenet datasets. For MAML, we use embedding function used by Vinyals et al. (2016), which has 4 modules with a $3 \times 3$ convolutions and 64 filters, followed by batch normalization, a ReLU nonlinearity, and $2 \times 2$ max-pooling. The Omniglot images are downsampled to $28 \times 28$, so the dimensionality of the last hidden layer is 64. As in the baseline classifier used by 2, the last layer is fed into a softmax. For Omniglot, we used strided convolutions instead of max-pooling. For ANIL, we only adjust the update strategy. For WCT, we use the same neural network architecture and learning configuration, with the except of the last layer (whose dimensions are the same as the number of categories). For MTL, we maintain the same setting with ANIL, except for bi-level optimization strategy.

**PsCo, Meta-GMVAE, UMTRA, and CACTUs.** We reuse the entire the configuration give by Jang et al. (2023a), Lee et al. (2021), Khodadadeh et al. (2019), Hsu et al. (2019), cause our test scenarios are the same as them.

**ReSSL and IIC** In CIFAR-10, CIFAR-100, STL-10, ImageNet, and Tiny ImageNet datasets, we reuse the entire the configuration describe in Zheng et al. (2022) and Ji et al. (2019). In DomainNet dataset, for a fair comparison, we use ResNet-9 as backbone and maintain the same learning configuration as mentioned above.

**MAE and NVAE.** Due to computational resource constraints, and in order to replicate the two approaches as much as possible, we use ViT-base (instead of ViT-Large) as backbone. In DomainNet dataset, we also use ResNet-9 as backbone. The learning configuration is in line with original.

**DeepCluster.** We run DeepCluster for each unsupervised zero-shot dataset, which we respectively randomly crop and resize to the appropriate image size. We modify the first layer of the AlexNet architecture used by the authors to accommodate this input size. We follow the authors and use the input to the (linear) output layer as the embedding. These are 4096-dimensional, so we follow the authors and apply PCA to reduce the dimensionality to 256, followed by whitening. Our configuration is built upon github.com/facebookresearch/ deepcluster. In DomainNet dataset, we also use ResNet-9 as backbone.

**BiGAN.** We follow the BiGAN authors and specify a uniform 50-dimensional prior on the unit hypercube for the latent. They use a 200 dimensional version of the same prior for their ImageNet experiments, so we follow suit for our unsupervised zero-shot dataset. They randomly crop to 64 × 64 and use the AlexNet-inspired architecture used by Donahue et al. (2017a) for their Imagenet results. Our configuration is built upon github.com/jeffdonahue/bigan. In DomainNet dataset, we also use ResNet-9 as backbone.

