# OpenReview forum: "Unsupervised Meta-Learning via Dynamic Head and Heterogeneous Task Construction for Few-Shot Classification"
_ICLR.cc/2025/Conference — ICLR 2025 Conference Withdrawn Submission_

### Official Review · Reviewer_3dPw · 2024-10-31

**Soundness:** 2
**Presentation:** 2
**Contribution:** 2
**Rating:** 3
**Confidence:** 5

**Summary:**

The authors of this paper present experimental results that demonstrate the advantages of the meta-learning method over the pre-trained method.

**Strengths:**

This paper suggests that the strength of meta-learning lies in its ability to handle heterogeneous tasks and label noise, potentially offering valuable insights for future research.

**Weaknesses:**

However, numerous comparisons between pre-trained models and meta-learning methods, both at theoretical and experimental levels, can be found in publications such as ICML and TPAMI. The method in this paper largely builds on existing approaches, with its main contribution limited to experimental analysis.

**Questions:**

1. Label noise is also a prominent topic in machine learning, and there are existing methods to address this issue in pre-trained models. Applying these methods could potentially further improve the results.
2. The comparison with state-of-the-art methods is not comprehensive, as it omits several important baselines, like, Revisiting Unsupervised Meta-Learning via the Characteristics of Few-Shot Tasks in TPAMI

---

### Official Review · Reviewer_9kpc · 2024-11-01

**Soundness:** 3
**Presentation:** 2
**Contribution:** 2
**Rating:** 3
**Confidence:** 4

**Summary:**

The paper aims to investigate meta-learning in the context of few-shot classification with questions as to why and when meta-learning performs well. In particular, the paper performs the analysis from the perspective of data noise and heterogeneous tasks. With analysis using Singular Vector Canonical Correlation Analysis, the paper claims that meta-learning algorithms are more robust to label noise and the task heterogeneity. Upon the analysis, the paper proposes a new framework for unsupervised meta-learning named DHM-UHT, that utilizes multi-head meta-learning with unsupervised heterogeneous task construction.

**Strengths:**

- The paper is easy to follow.
- The paper introduces a new framework that directly optimizes unsupervised heterogeneous task construction.
- The proposed method brings performance improvement.

**Weaknesses:**

- The paper seems to miss a detailed discussions on related works, especially on unsupervised meta-learning. The paper should include more detailed discussions on the similarities and differences from other works.

- There is another well-known multi-head meta-learning that can handle heterogeneous tasks, which is ProtoMAML [A]. How does the proposed method compare against ProtoMAML? The paper should include discussions with experimental comparisons against ProtoMAML used together with other unsupervised meta-learning algorithms, such as PsCo.

[A] Meta-Dataset: A Dataset of Datasets for Learning to Learn from Few Examples, ICLR 2020

- For more extensive experimental comparisons to strengthen the claim and the overall paper, the paper should include comparisons on more diverse datasets (such as the ones in [A]), under unsupervised few-shot scenario. This is because performance improvements on Omniglot and mini-imagenet seem marginal.

- The right figure in Figure 1 is a bit confusing. it's a bit hard to see whether it's a single output or multiple outputs, as they are put together  onto a single line.

- The paper does not follow standard ICLR submission format (no line number or ICLR submission header).

- The paper requires careful review: there are several typos and grammar errors, such as 'is is' at the end of page 6.

**Questions:**

I have incorporated questions in the weakness section.

---

### Official Review · Reviewer_kwEH · 2024-11-04

**Soundness:** 3
**Presentation:** 2
**Contribution:** 2
**Rating:** 5
**Confidence:** 5

**Summary:**

The authors investigate on the characteristic of meta-learning algorithms compared to conventional learning algorithms, where they introduce a previously known metric of "singular vector canonical correlation analysis (SVCCA)" to quantify the robustness of both learning algorithms. Additionally, the authors propose a new methodology based on their aforementioned findings where they employ dynamic head architecture and unsupervised task construction. They validate their approach on various datasets under few-shot scenarios.

**Strengths:**

- The motivation for the proposed method seems somewhat plausible and technically sound. With purpose of constructing a metric for evaluating the robustness and advantages of meta-learning algorithms compared to conventional learning algorithms.

- The proposed evaluation method seems to be generally applicable to few-shot learning methods to evaluate its robustness.

- The authors validate their approach with various meta learning approaches, on well-known few-shot learning datasets including miniImageNet and TieredImageNet, under various shot settings and cross-domain experiments.

**Weaknesses:**

- In the original literature that introduced SVCCA metric, it was used to compare the "similarity" between two representations across different layers and architectures. However, in terms of the proposed work, SVCCA was used to evaluate the "stability" of the learned representation across training epochs for a given network. Is it appropriate to compare SVCCA between representations across epochs to evaluate the "stability" of the learned representation? By looking at Figure 4 of the original SVCCA paper, it seems that comparing current representations vs. final representations seems more appropriate for evaluating the learning process and stability.

- Related to the previous question, in the original SVCCA literature, they also employed the freeze training method where lower layers are successively frozen during training. How does this apporach contribute to the "stability" of the learning process?

- Since the proposed algorithm is based on MAML, which inspired a large number of variants that employ bi-level optimization scheme for few-shot learning, proposed method should be tested on more baseline algorithms. Can the proposed algorithm achieve performance gains across other MAML-based algorithms?

- Minor points

  The manuscript is not properly formatted for review, without line numbers and proper page headers. Also, there are many sentences with typos and citations without brackets.

**Questions:**

Please refer to the questions in the weaknesses section.

---

### Official Review · Reviewer_eWZH · 2024-11-05

**Soundness:** 3
**Presentation:** 2
**Contribution:** 2
**Rating:** 3
**Confidence:** 4

**Summary:**

This paper mainly studies meta-learning in the problem of heterogeneous tasks and establishes a new algorithm namely DHM-UHT for dynamic head meta-learning. This paper focuses on the meta-learning robustness of label noise and the construction of heterogeneous tasks. The final experimental results were conducted on the zero-shot and few-shot datasets, indicating the proposed method achieves the state-of-the-art results.

**Strengths:**

+ Good experimental establishment and new evaluation metric to evaluate the performance namely SVCCA.
+ proposed the DHM-UHT achieves clear improvement on downstream datasets.

**Weaknesses:**

+ The key idea of SVCCA comes from existing literature, but is not novel or designed for this new problem. Besides, this new problem seems not well founded. The authors should provide more evidence to support the reviewers that this problem is distinctive and shows potential real-world usages.
+ For the second research, the meta-learning approach also comes from existing works, e.g., e Hospedales et al. (2021). The authors conduct simple improvements, e.g., the dynamic head, while focusing on a completely new problem. It is hard to convince the reviewers.
+ The presentations and formulations are not comprehensive and in good depth. The main formulation of this paper is to conduct several loss functions, including the cross-entropy.

**Questions:**

Please refer to the weakness section. It is hard for me to recognize why this problem is meaningful and the authors' proposed method seems to be a combination of the existing techniques.

**Details Of Ethics Concerns:**

The source code has been published in the prevailing link:https://github.com/tuantuange/DHM-UHT
I really appreciate the authors releasing their source code.


But this link might not follow the double-blind reviewing policy of the ICLR confernece. I report this problem and suggest the ethics reviewers involve in this issue.

---

### Note · Authors · 2024-11-13

I have read and agree with the venue's withdrawal policy on behalf of myself and my co-authors.